Molecular Biology and Physiology
# Chaperone-Mediated Stress Sensing in *Mycobacterium tuberculosis* Enables Fast Activation and Sustained Response

Satyajit D. Rao,[a] Pratik Datta,[b] Maria Laura Gennaro,[b] Oleg A. Igoshin[a]

aDepartment of Bioengineering and Center for Theoretical Biological Physics, Rice University, Houston, Texas, USA
bPublic Health Research Institute, New Jersey Medical School, Newark, New Jersey, USA

Satyajit D. Rao and Pratik Datta contributed equally to this work; author order was decided based on the contribution to manuscript preparation.

**ABSTRACT** Dynamical properties of gene regulatory networks are tuned to ensure bacterial survival. In mycobacteria, the MprAB-$\sigma^E$ network responds to the presence of stressors, such as surfactants that cause surface stress. Positive feedback loops in this network were previously predicted to cause hysteresis, i.e., different responses to identical stressor levels for prestressed and unstressed cells. Here, we show that hysteresis does not occur in nonpathogenic *Mycobacterium smegmatis* but does occur in *Mycobacterium tuberculosis*. However, the observed rapid temporal response in *M. tuberculosis* is inconsistent with the model predictions. To reconcile these observations, we implement a recently proposed mechanism for stress sensing, namely, the release of MprB from the inhibitory complex with the chaperone DnaK upon the stress exposure. Using modeling and parameter fitting, we demonstrate that this mechanism can accurately describe the experimental observations. Furthermore, we predict perturbations in DnaK expression that can strongly affect dynamical properties. Experiments with these perturbations agree with model predictions, confirming the role of DnaK in fast and sustained response.

**IMPORTANCE** Gene regulatory networks controlling stress response in mycobacterial species have been linked to persistence switches that enable bacterial dormancy within a host. However, the mechanistic basis of switching and stress sensing is not fully understood. In this paper, combining quantitative experiments and mathematical modeling, we uncover how interactions between two master regulators of stress response—the MprAB two-component system (TCS) and the alternative sigma factor $\sigma^E$—shape the dynamical properties of the surface stress network. The result show hysteresis (history dependence) in the response of the pathogenic bacterium *M. tuberculosis* to surface stress and lack of hysteresis in nonpathogenic *M. smegmatis*. Furthermore, to resolve the apparent contradiction between the existence of hysteresis and fast activation of the response, we utilize a recently proposed role of chaperone DnaK in stress sensing. These result leads to a novel system-level understanding of bacterial stress response dynamics.

**KEYWORDS** *Mycobacterium tuberculosis*, chaperones, mathematical modeling, sigma factors, stress response, two-component regulatory systems

The intracellular pathogen *Mycobacterium tuberculosis* is highly successful in humans, as more than a billion individuals are estimated to carry a latent infection. To survive in the host, *M. tuberculosis* must sense stress conditions generated by the host immune system and adapt to them by reprogramming its gene expression and metabolism. Cell envelope damage is one such stress condition (1, 2). As in many bacteria, the response to this stress involves a complex gene regulatory network involving transcriptional master regulators, namely, two-component systems (TCSs) and alternative sigma factors (2–4).

Address correspondence to Oleg A. Igoshin, igoshin@rice.edu.

Hysteresis in Mycobacterium tuberculosis stress-response network is uncovered by a combination of mathematical modeling and quantitative experiments.

The core surface stress response network of *M. tuberculosis* involves the alternative sigma factor $\sigma^E$ and the MprAB TCS, which consists of a histidine kinase (MprB) and a response regulator (MprA). The presence of surface stressors (surfactants) triggers autophosphorylation of MprB at the histidine residue. The phosphoryl group is then transferred to the Asp residue of MprA. MprB also has phosphatase activity, whereby it catalytically dephosphorylates phosphorylated MprA (MprA-P) (5, 6). MprA-P is a transcription factor that activates expression of multiple genes, including that of its own operon, thus creating a positive feedback loop. Another transcriptional target of MprA is *sigE*, the gene encoding $\sigma^E$ (7). This alternative sigma factor binds core RNA polymerase and guides it to promoters of stress-responsive genes, including the *mprA-mprB* operon. Upregulation of *mprA-mprB* by $\sigma^E$ results in a second positive feedback loop (4, 8). In addition, $\sigma^E$ activity is also controlled posttranslationally by sequestration by the anti-sigma factor RseA, which disables $\sigma^E$ from interacting with RNA polymerase [8, 9].

The MprAB-$\sigma^E$ stress response network has recently been linked with persistence, a state in which tubercle bacilli survive inside host immune cells, where they encounter nutrient and oxygen limitation and antibacterial mechanisms (10–12). The mechanism (s) behind cells switching from bacterial growth to a persistent state remains unknown. However, a target of $\sigma^E$, *relA* (a stringent response regulator), showed a bimodal distribution in a population of *Mycobacterium smegmatis* cells (13, 14). A bimodal distribution in a gene's expression level can arise out of bistability in the MprAB-$\sigma^E$ network (i.e., the existence of two distinct states of response for the same level of stress), and a bistable network is a good candidate for a persistence switch if one of the states ceases growth. This possibility was explored by a previous theoretical study from our team (15). Its results demonstrated that positive feedbacks in this network, together with increased effective cooperativity due to RseA-$\sigma^E$ interaction, could result in bistability over a wide range of parameter values. As a result, for a certain range of signals, the transcription activity of $\sigma^E$ can be either high (activated) or low (inactivated), depending on initial conditions (see Fig. 5 in reference 15). Bistability would then manifest as hysteresis in response to increasing and decreasing signals, i.e., fully prestressed and unstressed cells may show different $\sigma^E$ activity under identical intermediate stress levels. However, this theoretical prediction of bistability has not been experimentally tested.

Here, we investigate whether the predicted hysteresis in the transcription activity of $\sigma^E$ is observed experimentally in two mycobacterial species, the nonpathogenic *M. smegmatis* and pathogenic *M. tuberculosis*. To investigate the possibility of hysteresis, we examine mycobacterial response to increasing and decreasing surface stress created by different concentrations of surfactant SDS. Furthermore, we compare model predictions of transient activation or deactivation of $\sigma^E$ activity following addition or removal of SDS. Using mathematical modeling and parameter fitting, we identify interactions in the network that explain experimentally observed responses. We find that a simple model assuming a first-order activation of MprB autophosphorylation in response to SDS exposure fails to explain the experimental observations. To resolve the inconsistency, we propose a more complex model for MprB activation. This model implements a proposed mechanism involving DnaK, a chaperone that deactivates MprB in unstressed conditions (16). Extracytoplasmic proteins unfolded due to surfactant exposure compete with MprB for DnaK, eventually activating the MprAB TCS. The model not only explains the observed dynamical properties of the stress response but also predicts changes in stress response with perturbed DnaK levels. These predictions are confirmed with an engineered strain, confirming the assumptions of the model. Thus, synergistic use of experiments and modeling can uncover interactions in signaling networks that shape their dynamical properties.

## RESULTS

**Lack of hysteresis in transcript levels in *M. smegmatis* can be explained by lack of positive feedback.** To experimentally investigate the possibility of bistability, we first examine the response of *M. smegmatis* to increasing and decreasing levels of surfactant SDS that causes cell envelope stress. If our prediction of bistability in this

network is correct, we expect to observe hysteresis, i.e., different intermediate states depending on cell history. To test this possibility, previously unstressed or maximally stressed cells were subjected to different intermediate concentrations of SDS. Maximal stress was first identified as the concentration of SDS resulting in bacteriostasis (0.02%; see Fig. S1 in the supplemental material). The state of the network was measured as *sigE* transcript abundance following SDS exposure for 2 h. The results show no hysteresis in *sigE* over the range of SDS concentrations (Fig. 1a).

Given that positive feedback is required for bistability, we test whether this feedback exists in *M. smegmatis*. We measured *mprA* transcript abundance in wild-type and *sigE*-deletion strains at regular time intervals following exposure to maximal SDS = 0.02% (reffig: exptb). At all time points, the transcript abundance of *mprA* in the *sigE* deletion mutant does not deviate significantly from that in the wild type. These measurements indicate that *sigE* deletion has no measurable effect on *mprA* transcription during SDS stress. Additionally, bioinformatics search for a $\sigma^E$ binding site using a consensus sequence described previously (4) in the upstream region of *mprA* did not yield any matches. In contrast, *sigE* expression is highly attenuated compared to the wild type in an *mprA* deletion strain (Fig. 1c). Together, our data show that MprA activates *sigE* transcription but that the feedback loop from $\sigma^E$ to *mprA* is absent in *M. smegmatis*. Thus, lack of bistability, and therefore hysteresis, could be due to a lack of positive feedback.

**Hysteresis observed in *sigE* transcript levels in *M. tuberculosis*.** In contrast to the *M. smegmatis* data presented above, reports have suggested that the feedback from $\sigma^E$ to MprA exists in *M. tuberculosis* (7, 8). Here, we confirm these observations, demonstrating that transcription of *mprA*, following exposure of *M. tuberculosis* cells to SDS, is significantly decreased in a *sigE* deletion mutant in comparison to that in wild-type cells (Fig. 1e). With the existence of two positive feedback loops confirmed (8, 17), we examined whether hysteresis could be observed in *M. tuberculosis*. We measured *sigE* transcripts in previously unstressed and maximally stressed cells exposed to intermediate concentrations of SDS for 2 h. For maximal stress, bacteriostatic SDS at 0.03% was used (18). The results indicate that, unlike that of *M. smegmatis*, the *M. tuberculosis* network exhibits hysteresis (Fig. 1d). Taken together, our findings in *M. tuberculosis* and *M. smegmatis* strongly suggest a role for positive feedback in hysteresis in mycobacterial response to surface stress.

Notably, in prestressed cells, *sigE* transcript levels remain above basal levels even after the removal of stressors, i.e., at 0% of SDS (Fig. 1d). This is in contrast to the predictions of our previous model in Tiwari et al. (15). To illustrate this, we have simulated the dose-response relationship using this model and parameters (Fig. 2a). Using MprB autophosphorylation as the signal mimicking the surface stress, we computed the steady-state *sigE* mRNA level as a function of signal. Black and red curves correspond to different initial conditions; the system can start at the steady-state concentrations corresponding to low (black) and high (red) signals, respectively (i.e., initially unstressed or maximally stressed cells). The results show that while different steady states are predicted to occur at the intermediate range of signal, *sigE* transcripts return to basal levels when signal is low. This is in contrast to the observations in Fig. 1d. Therefore, the mechanism of hysteresis may be more complicated than the previous model suggested, or the model operates in the wrong parameter regime.

**Hysteresis is seen in both *mprA* and *sigE*, but the dynamics are unexpectedly fast.** To further compare the predictions of our previous model (15) to experimental observations, we test the prediction of hysteresis in *mprA* mRNA (Fig. 2b) by measuring *mprA* transcripts in previously unstressed and maximally stressed cells exposed to intermediate SDS concentrations for 2 h. The results show that *mprA* transcripts display hysteresis (Fig. 3c; triangles). We note that, in contrast with *sigE* and in agreement with model prediction, *mprA* transcripts reach basal levels after removal of SDS (Fig. 3c).

In addition to testing model predictions of steady-state responses, we investigated the response dynamics following activating and deactivating signals. Notably, time course simulation of our previous model (15) shows very slow response (Fig. 2c and d). Starting with an initial condition corresponding to low signal (i.e., unstressed

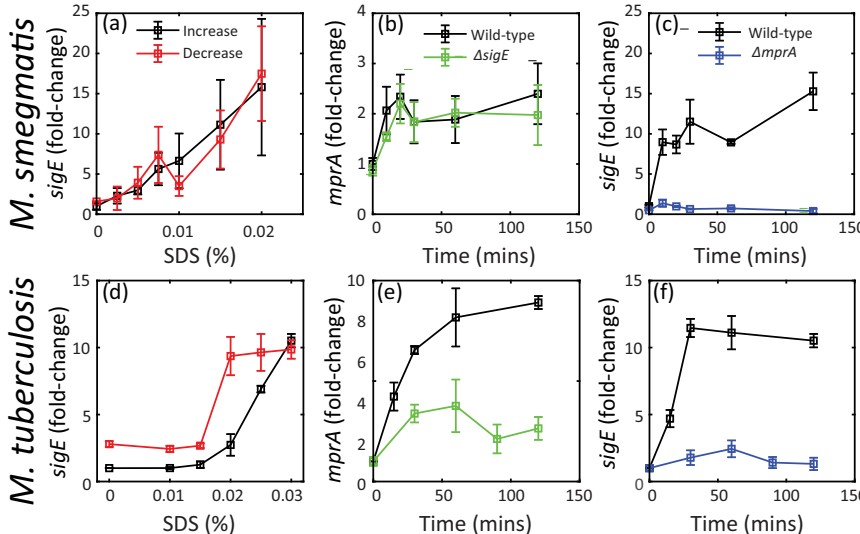

**FIG 1** Dynamics of stress response in *Mycobacterium smegmatis*. Bacterial transcripts were enumerated by real-time PCR using gene specific probes. Transcripts were normalized to 16S rRNA and expressed as fold change relative to pretreatment. Here and in subsequent figures, mean values (± standard error of the mean) are presented from triplicate experiments. (a) *M. smegmatis* does not display dose-response hysteresis in mRNA levels. Wild-type cells were grown up to the mid-log phase and treated with increasing SDS concentrations and harvested 2 h posttreatment (black). Wild-type cells were treated with bacteriostatic SDS (0.02%; see Fig. S1 in the supplemental material) for 2 h, centrifuged, and resuspended in fresh medium containing the same or decreasing SDS concentrations and harvested 2 h posttreatment (red). (b) Deletion of the *sigE* gene does not affect the *mprA* time course, suggesting that *M. smegmatis* lacks feedback from $\sigma^E$ upregulating *mprA*. Mid-log cultures of the wild type and an *sigE* deletion mutant were treated with 0.02% SDS and harvested pretreatment (time 0) and at multiple times posttreatment. (c and f) Similar time course measurement of *sigE* mRNA in the wild type and an *mprA* deletion mutant shows very low fold change in expression, suggesting that MprA regulates *sigE* expression in both *M. smegmatis* and *Mycobacterium tuberculosis*. (d) *M. tuberculosis* dose-response displays hysteresis in mRNA levels. (e) Deletion of *sigE* gene affects the *mprA* time course, suggesting that *M. tuberculosis* has feedback from $\sigma^E$ upregulating *mprA*.

condition), we simulate exposure to maximal stress level by stepwise change in MprB autophosphorylation rate to the value corresponding to saturated steady-state response (1 s$^{-1}$). The results show unrealistically slow kinetics of mRNA accumulation (Fig. 2c and d), with a predicted response time on the order of ~33 h. If that is true, the protocol used to measure hysteretic response may not be sufficient to achieve steady state. We note that

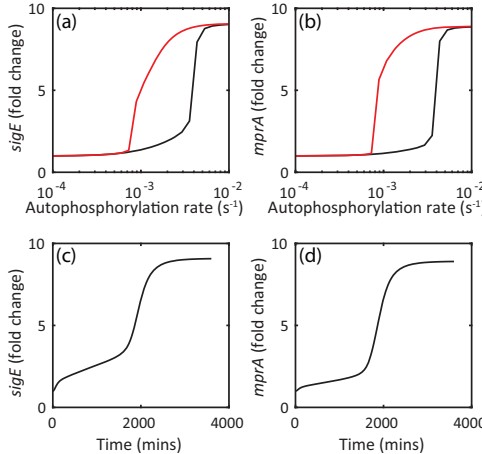

**FIG 2** (a and b) Steady-state dose-response simulations of *mprA* and *sigE* starting from low (black) and high (red) signal predict hysteresis. (c and d) A very slow response (on the order of 2,000 min) is predicted for a switch from low to high signal.

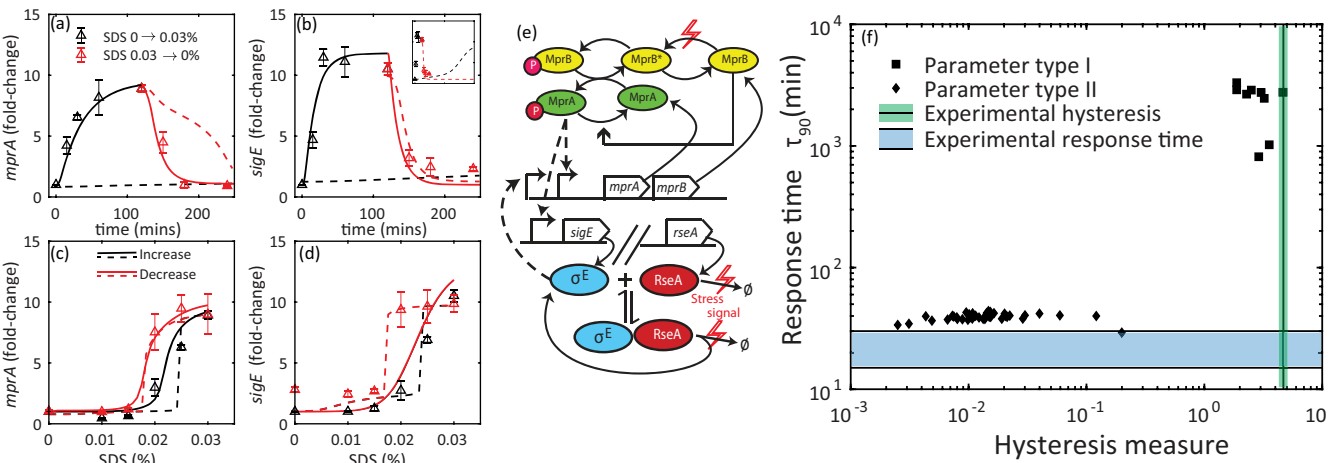

**FIG 3** Hysteresis and response dynamics of *M. tuberculosis* response to SDS stress. (a to d, triangles) Dose-response and time course enumeration of transcripts in *M. tuberculosis*. (a and b) Time course measurements show rapid accumulation of transcripts. See "Experimental methods" for protocols. Cultures were exposed to 0.03% SDS, and *sigE* (MprA-P target) and *mprA* ($\sigma^E$ and MprA-P target) mRNA levels were enumerated at multiple time points over the course of 2 h (black triangles). Measurements are presented here as fold change with respect to pretreatment levels. Cultures were subsequently resuspended in fresh SDS-free medium to measure a step-down time course (red triangles). (c and d) Dose-response measurements show hysteresis in *sigE* and *mprA* mRNA levels depending on history of stress exposure, namely, previously stressed (red triangles) and unstressed (black triangles). (e) Schematic of the MprAB-$\sigma$E network model. Red lightning symbols represent signal (SDS exposure). SDS exposure leads to MprB switching from phosphatase to kinase form, and also to degradation of RseA (Ø indicates RseA degradation products) and release of $\sigma^E$. Dashed arrows indicate transcriptional activation. (a to d) Solid lines show fitting parameters for a simple model of two-component system (TCS) activation, as depicted in (e). The resulting simulations match time course data well (a and b), but dose-response simulation of *sigE* mRNA shows no hysteresis (c and d). In models that minimize steady-state dose-response (c and d, dashed black lines), time course simulation suffers a large activation delay (a and b, dashed black lines). Inset shows activation time on the order of 1,000 min. Simulated mRNA levels were normalized to respective pretreatment values. (f) Analysis of multiple parameter sets that generate fits of the type shown in panels a to d illustrates the trade-off between explaining fast response time (diamonds) and hysteresis (squares) in the simple model of MprAB TCS activation shown in panel a. Shaded regions indicate response time and a measure of hysteresis ranges (see Materials and Methods) computed from experimental data.

predicted slow response is consistent with expectation of critical slowdown in kinetics around a bistable threshold (19). To experimentally test this prediction, we exposed previously unstressed cells to 0.03% SDS and measured *sigE* and *mprA* transcripts at regular intervals for 2 h following treatment (see Materials and Methods). The results (Fig. 3a and b; black triangles) show that both *mprA* and *sigE* transcripts accumulate quite rapidly in contrast to model predictions. Thus, the discrepancy between the observed rate of transcript accumulation and the predictions of our previous model (15) requires us to revisit our network model and parameters.

**Models with simple activation mechanism of the MprAB two-component system cannot explain dynamical properties.** To understand the mechanisms that lead to unexpectedly fast accumulation of target mRNAs, we start with our previous model (15), with two slight modifications in order to account for two important but previously unaccounted aspects of stress-sensing mechanisms (Fig. 3e). First, instead of modeling stress by increasing the MprB autophosphorylation rate, we assume that stress controls both kinase and phosphatase activity of MprB. In many two-component systems, sensory transduction is driven by a conformation change, enhancing kinase and decreasing phosphatase activity (20–22). Thus, we include two conformations of MprB explicitly in the model. In the absence of surface stress, MprB is phosphatase dominant (16) and switches to a kinase form in response to stress. The stress signal modulates the first-order rate constant of switching between these two forms. Second, we explicitly introduce stress-dependent modulation of RseA activity. In the presence of SDS, RseA has been reported to undergo phosphorylation by a transmembrane kinase, PknB, and subsequent proteolytic degradation (9). We include this in the model by introducing an additional RseA degradation reaction with a signal-dependent rate constant (see Materials and Methods).

With the revised model, we seek to generate parameters that minimize deviations

of simulated time course and the dose-response relationship from experimentally observed data points. To this end, we employ minimization of the sum-of-squares errors between the model predictions and experimental data points using particle swarm optimization (see Materials and Methods). We seek to obtain a large number of parameter sets in this way to account for some parameters having more or less effect on the measured variables. None of the optimized parameters sets is adequate to explain both steady-state and dynamic response (Fig. 3a to d).

To understand the reasons for this discrepancy, we focus on separately optimizing these data sets. While simulations using optimized parameters can match the time course of mRNA accumulation (Fig. 3a and b; solid lines), they lack dose-response hysteresis, especially in *sigE* (Fig. 3c and d; solid lines). At intermediate SDS, the simulated *sigE* mRNA levels are the same regardless of the initial condition of the network—OFF or ON (Fig. 3d; black and red lines overlap). Given that our previous analysis suggested that the MprAB-$\sigma^E$ network can be bistable (15), we attempt to match only dose-response data points at steady state by relaxing the condition for rapid mRNA accumulation. Parameter sets that minimize only steady-state dose-response error show a close match with experimental data (Fig. 3c and d; dashed lines). However, a simulated time course of mRNA accumulation shows a much slower response than the experimental data (Fig. 3a and b, dashed black lines, and Fig. 3b, inset). This suggests that, while the network model can match time course and dose-response experimental data points separately, it is unable to do so when the two data sets are included simultaneously (discussion for this trade-off follows in the next section). While the simulations in Fig. 3 are representative, the trade-off between hysteresis and fast accumulation time is robust. We illustrate this with a scatterplot of a measure for hysteresis and response time (Fig. 3f; see also Materials and Methods). Each point represents a parameter set that fits one data set adequately (either time course [diamonds] or steady-state dose-response [squares]), obtained from runs of particle swarm optimization with random initial seeds. No optimized parameter sets occupy the space at the intersection of experimental hysteresis and response time measures (shaded areas).

**Robustness property of TCSs may lead to a trade-off between hysteresis and response speed in a simple TCS model.** To understand why hysteresis is absent when using parameter sets that match the time course (Fig. 3a to d, solid lines), we analyzed our ordinary differential equation (ODE) model with time scale-separated modules as described previously (15). We find that the MprAB two-component system lies in a regime of absolute concentration robustness that has been observed previously in TCSs in bacterial systems (see Fig. S2 in the supplemental material) (23–25). In this regime, the output of a TCS (MprA-P) is invariant to total MprA/MprB concentrations. This in turn ensures that MprA-P output (*sigE* mRNA) is invariant to positive feedback. In contrast, with parameters that describe hysteresis but not rapid accumulation (Fig. 3a to d, dotted lines), the TCS lies outside this regime where the MprA-P depends on the total amount of MprA present in cells (Fig. S2; dotted lines). In fact, in this regime, MprA is saturated, and almost all of it is phosphorylated. The modules intersect in 3 points, showing bistability in the network with parameters that display hysteresis (the intermediate intersection represents unstable steady state). The steady state of the network would depend on the initial condition. Thus, we conclude that a rapidly activating TCS cannot display dose-response hysteresis due to robustness properties of two-component systems. Conversely, models displaying hysteresis cannot obtain rapid activation dynamics due to being close to bistable threshold (26).

**DnaK-dependent activation of MprB resolves the trade-off between response speed and hysteresis.** To resolve the previously described trade-off, we look for network designs that can generate bistability in a biological network without activation delays. A potential design consists of a transcription factor (TF) sequestered by a stoichiometric inhibitor, coupled with positive autoregulation of TF (19). In the absence of activating signal, inhibitor concentration exceeds that of TF, keeping it inactive. If activating signals titrate the inhibitors, TF is released and can upregulate its transcriptional target. When positive feedback is present in the network, prolonged exposure to signal

can lead to accumulation of TF to levels exceeding those of the inhibitor. In that case, there can be a residual TF activity even in the absence of signal. Furthermore, given that activation is driven by posttranslational sequestration reactions, its dynamics should be fast. In fact, such activation mechanisms appear plausible in the MprAB-$\sigma^E$ network. DnaK, a mycobacterial chaperone protein, has been shown to bind to the extracytoplasmic domain of MprB and suppress its autokinase activity (16). In an *M. tuberculosis* mutant strain expressing *dnaK* from a chemically inducible promoter, the MprAB-$\sigma^E$ network did not activate even after exposure to 0.05% SDS (16), suggesting that the concentration of DnaK is an important factor for stress response activation. We implement the following mechanism for activation of MprAB based on the results of Bretl et al. (16). MprB can only autophosphorylate when not bound to DnaK and only has phosphatase activity when bound to DnaK. Exposure to SDS increases the load of unfolded/misfolded extracytoplasmic proteins. Recruitment of chaperone DnaK to those proteins releases MprB to autophosphorylate and activate MprA.

To test whether this modified MprAB TCS can match the experimentally observed hysteresis, we incorporate MprB-DnaK binding in our previous model (Fig. 4a). Instead of representing surface stress by a single kinetic rate constant as in the previous model, we model exposure to SDS as a step increase in a hypothetical DnaK target representing misfolded/unfolded extracytoplasmic proteins (see Materials and Methods for more details). This increase consequently reduces the concentration of DnaK available to bind MprB. When implemented, we use this MprAB-DnaK-$\sigma^E$ model to fit the measured experimental data. As a result, we are able to obtain multiple parameter sets with which the model exhibits hysteresis at intermediate SDS concentrations (Fig. 4b to e) and matches the observed activation kinetics. Bifurcation analysis of the model as a function of signal (SDS concentration) shows that the model is indeed bistable at intermediate signal levels (see Fig. S3 in the supplemental material). Notably, when we adapt the MprAB-DnaK-$\sigma^E$ model to fit *M. smegmatis* data by removing positive feedback from $\sigma^E$ to MprAB, we can reconcile the absence of hysteresis in the dose-response relationship (see Fig. S4 in the supplemental material). Using the model and parameters obtained for *M. tuberculosis*, however, we find that the models fail to reproduce residual transcriptional activity in prestressed cells for the low SDS concentrations (less than 0.015%) observed in the data (Fig. 4e; compare red line with red triangles). By repeating the experiment on 3 different days, we demonstrated that residual levels of *sigE* mRNA even 2 h after complete SDS removal are highly reproducible (Fig. 4f). Therefore, we conclude that further modifications of the proposed model are needed to fully explain the data. In our model, only phosphorylated MprA is transcriptionally active. Thus, despite accumulation of MprA due to positive feedback, rapid dephosphorylation of MprA-P following SDS removal leads to negligible *sigE* transcription.

Given *in vitro* studies showing that unphosphorylated MprA can also bind MprA-P target promoter DNA (7), we hypothesize that unphosphorylated MprA might act as a weak (i.e., with lower promoter affinity) transcriptional activator of *sigE* (Fig. 4g). Addition of this interaction to the model (see Materials and Methods) can explain this residual transcription activity, leading to a better fit overall (Fig. 4h and i). Thus, the trade-off between hysteresis and response time is resolved. We illustrate this with a scatterplot of hysteresis measures and response times for best-fitting parameter sets obtained from multiple particle swarm optimization runs (see Fig. S5 in the supplemental material). The points occupy the space at the intersection of experimental hysteresis and response time left unoccupied by the previous model (Fig. 3f and Fig. S5).

**Overexpressing DnaK from the stress-responsive promoter partially abrogates stress activation.** If the hypothesized mechanism of hysteresis is accurate, we expect the dynamical properties of the network (activation level and hysteresis) to be strongly sensitive to DnaK concentration. The *dnaK* gene is essential for growth in *M. tuberculosis*, and therefore deletion mutants are not viable (27, 28). Severe overexpression of

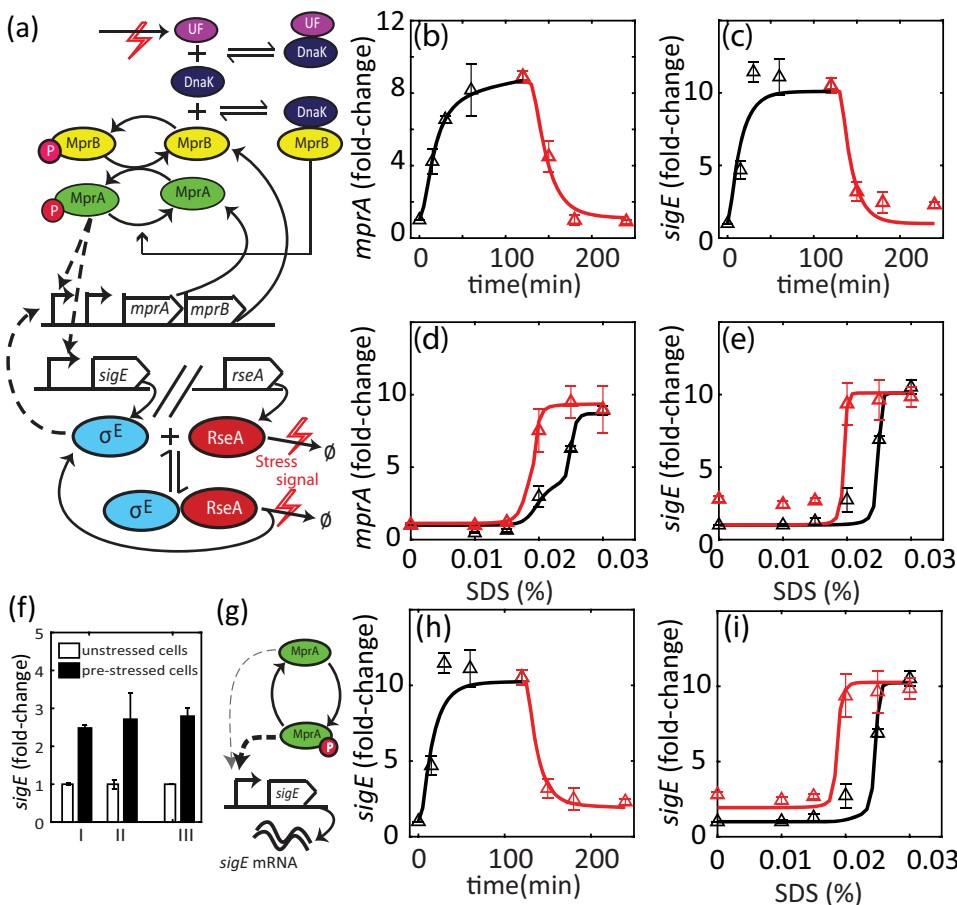

**FIG 4** (a) Modification to MprA/B TCS. Chaperone DnaK binds MprB and suppresses autokinase activity. Processes affected by SDS exposure are denoted by a red lightning bolt symbol. SDS leads to build up of unfolded/misfolded extracytoplasmic proteins (unfolded protein load [UF]), which are targets of DnaK. DnaK subsequently releases MprB, which is then free to autophosphorylate and catalyze phosphotransfer to MprA, thus activating the TCS. SDS exposure also leads to degradation of RseA (Ø indicates degradation products) and release of $\sigma^E$. Dashed lines indicate transcriptional regulation. (b to d) Parameter fitting to model with modified TCS. This model can simulate close matches to the time course (b and c). Moreover, dose-response simulations show hysteresis depending on history of SDS exposure (d and e). However, residual *sigE* mRNA (e) (red triangles) cannot be explained by this model. (f) Residual activity of *sigE* can be observed in multiple biological replicates. Measurement of *sigE* in the prestressed cells at 2 h after removing SDS (solid bars) shows nearly 2.5-fold more *sigE* mRNA than that in the unstressed cells (hollow bars). (g) Unphosphorylated MprA binds the *sigE* promoter (with lower affinity than MprA-P). (h and i) Residual *sigE* activity can be explained by this hypothesis.

DnaK has already been shown to abrogate induction of *sigE* and *mprA* mRNA following exposure to SDS (16). Here, we test the effects of DnaK overexpression on dose-response hysteresis. To perturb DnaK expression levels, we integrate into the genome an extra copy of *dnaK* expressed from the *mprA* promoter (Fig. 5a; see Methods and Methods). In this engineered strain, the native copy of *dnaK* expresses the chaperone constitutively, whereas the additional copy expresses it in a stress-dependent manner (Fig. 5a). We argue that basal level of this promoter (in the absence of stress) may not be sufficient to fully attenuate the stress response.

With the engineered strain, we conduct dose-response experiments as described previously. We find that overexpressing DnaK results in attenuated activation of target genes (Fig. 5b and c). While the transcript levels of *mprA* increase nearly 3-fold over the range of SDS concentrations, *sigE* mRNA is not increased significantly. Since the *sigE* promoter is activated by MprA-P, this suggests that MprAB TCS is not activated follow-ing SDS treatment. On the other hand, modest upregulation of *mprA* could indicate

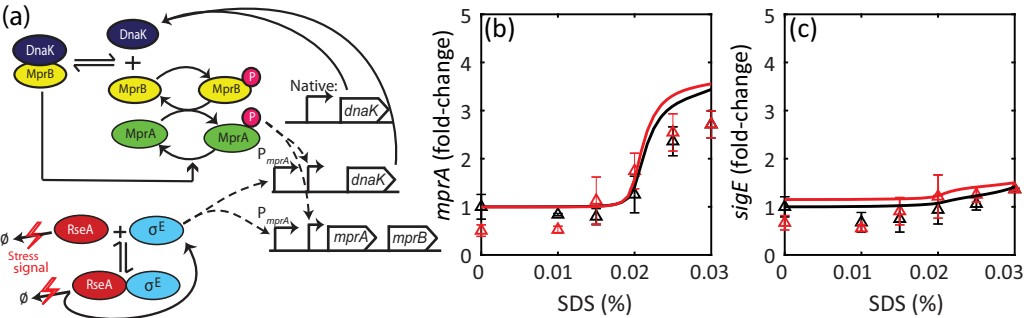

**FIG 5** DnaK overexpressed from stress-responsive promoter. (a) Schematic of MprAB-DnaK for the strain expressing DnaK from the *mprA* promoter (PmprA) in addition to the native copy. (b and c) Much lower fold activation of *sigE* and *mprA* transcripts compared to the wild type. DnaK-overexpressing cells were grown up to the mid-log phase, treated with increasing SDS concentrations, and harvested 2 h posttreatment (black triangles). DnaK-overexpressing cells were treated with bacteriostatic SDS (0.03%) for 2 h, centrifuged, and resuspended in fresh medium containing the same or decreasing SDS concentrations and harvested 2 h posttreatment (red triangles). Solid lines show model simulations with DnaK translated from additional mRNA equal in concentration to *mprAB* mRNA (see Materials and Methods).

activation of the MprAB-$\sigma^E$ network through the alternative independent pathway of RseA degradation to release $\sigma^E$. Our modeling results are consistent with this hypothesis. When we simulate the stress response of the engineered strain (see Materials and Methods), it is possible to obtain a qualitatively consistent transcript dose-response relationship by tuning *dnaK* translation rate as a free parameter (Fig. 5b and c). Since $\sigma^E$-RseA interaction in our model is completely independent of DnaK, we find that $\sigma^E$ activates the *mprA* promoter, leading to a modest upregulation. Induction of *mprA* is eliminated in the model in which no stress-dependent RseA degradation is present or if $\sigma^E$ is knocked out (see Fig. S6 in the supplemental material). In these simulations, even the basal concentration of DnaK expressed from the extra copy of *dnaK* is nevertheless sufficient to inhibit MprB, which is also expressed at the basal level. Following SDS exposure, no significant upregulation of either is observed due to the loss of $\sigma^E$-dependent activation of the *mprA* promoter driving the extra copy of *dnaK*.

We note that the experimentally observed change in stress response dynamics is not due to decrease in viability of engineered strain in SDS. Despite attenuation of the stress response, at maximal SDS concentration used for hysteresis measurements (bacteriostatic concentration, 0.03%), cells remain viable over the experimental time frame, as shown by survival curves (see Fig. S7 in the supplemental material).

Taken together, these results suggest that increasing DnaK levels in *M. tuberculosis* leads to partial abrogation of the stress response and suggest DnaK-dependent activation of the MprAB TCS and DnaK-independent activation of $\sigma^E$.

## DISCUSSION

The surface stress-responsive TCS MprAB, together with the alternative sigma factor $\sigma^E$, forms a stress response network in *M. tuberculosis* that is a viable candidate for a bistable switch. Here, we test the predicted bistable switch in the MprAB-$\sigma^E$ network in mycobacterial strains by measuring gene expression in response to stress. Consistent with our previous prediction of bistability (15), we found that previously stressed cells show significantly higher levels of stress-activated transcripts compared to previously unstressed cells exposed to the same concentration of SDS. However, in contrast to predictions from our previous bistable model, we observed rapid accumulation of transcripts, suggesting that the assumed signaling network is inconsistent with experimentally observed dynamical properties. Our finding of a trade-off between hysteresis and response time in this model of the MprAB-$\sigma^E$ network explains this inconsistency. We propose that the recently suggested mechanism (16) for activation of MprAB mediated by the chaperone DnaK can lead to bistability. Crucially, this mechanism does not result in a trade-off and can explain all experimental observations. Furthermore, our model predictions of the effects of DnaK

perturbation are consistent with experimental measurements of engineered strains of *M. tuberculosis*.

We find that *M. smegmatis* neither displays hysteresis nor has a second positive feedback loop by which $\sigma^E$ regulates the *mprAB* operon. This result indicates that hysteresis is linked to mutual activation between MprAB and $\sigma^E$. Since signaling architectures typically evolve in response to the requirement to survive in stressful environments (29–31) and *M. smegmatis* is a nonpathogenic strain, it is tempting to speculate that the dynamical properties gained from the double positive feedback loop architecture in *M. tuberculosis* might be necessary for virulence (32, 33).

In modeling of two-component systems with a bifunctional kinase, it is commonly assumed that the activating signal simply increases the autophosphorylation rate, thereby decreasing the fraction of the unphosphorylated kinase that can act as phosphatase. This assumption has been used in numerous modeling and theoretical analyses and has been sufficient to explain many observed dynamical properties of bacterial TCSs (23, 34–36). In our previous study predicting bistability in this network (15), the autokinase rate of MprB was assumed to increase with stress. In contrast, in this study guided by the constraints set by our time course and dose-response measurements, we found that such an activation assumption led to a trade-off between hysteresis and response time. The trade-off is resolved by a more detailed activation mechanism involving the chaperone DnaK. Notably, the presence of a third protein stoichiometrically interacting with the kinase will make systems response sensitive to changes in the concentrations of the two components. This is in contrast with the absolute concentration robustness regime when a third component is lacking (24, 25). It is interesting to see how potential loss of fitness due to lack of robustness in the DnaK-dependent activation mechanism may be compensated with fitness advantage due to the fast and sustained response of a bistable network. Arguably, the latter might be more important for virulent mycobacteria.

Involvement of the chaperone DnaK with TCS signaling has precedents, since an effector protein regulating the activity of envelope stress sensor kinase has been observed in other bacterial species. The response to envelope stress in *Escherichia coli* is controlled by the CpxAR TCS. A third component, CpxP, interacts with CpxA and suppresses its kinase activity in the absence of stress (37). Under conditions leading to overexpression of misfolded envelope proteins, CpxP is recruited away from CpxA, thus activating the TCS (38, 39). Overexpressing CpxP results in reduced Cpx response (38, 39). A similar mechanism exists in mammalian unfolded protein responses, where BiP acts as a folding chaperone for misfolded peptides exiting the endoplasmic reticulum (ER). In addition to its role as a chaperone, BiP also binds and negatively regulates the activities of three transmembrane ER stress transducers, PERK, ATF6, and IRE1 (40). These three signaling proteins are released under conditions of increased load of misfolded peptides in the ER. Interestingly, overexpression of BiP leads to reduced activation of IRE1 and PERK. This could suggest a mechanism for detecting misfolded protein loads that allows for a rapid activation of stress response while sustaining activity, even as the stress decreases. Therefore, our discovery of the dynamical consequences of a chaperone-mediated activation of signaling can help understand stress-response and protein homeostasis in diverse organisms.

## MATERIALS AND METHODS

**Experimental methods. (i) Bacterial strains, reagents, and growth conditions.** *M. tuberculosis* H37Rv, *Mycobacterium smegmatis* (Mc2 155), and *Escherichia coli* XL1 blue (Agilent Technologies, Santa Clara, CA) were used. *M. tuberculosis* were grown in Middlebrook 7H9 broth (liquid) and Middlebrook 7H10 agar (solid) (Difco, Franklin Lakes, NJ), supplemented with 0.05% Tween 80, 0.2% glycerol, and 10% ADN (2% glucose, 5% bovine serum albumin, and 0.15 M NaCl). However, 10% ADN was excluded from the 7H9 and 7H10 media while *M. smegmatis* were grown. For DNA cloning, *Escherichia coli* XL1 blue (Agilent Technologies) was grown in Luria-Bertani (LB) broth or agar (Thermo Fisher Scientific, Waltham, MA). As needed, solid and liquid media were supplemented with 25 or 50 $\mu$g · ml$^{-1}$ kanamycin sulfate (Thermo Fisher Scientific) for *Mycobacterium* spp. and *E. coli*, respectively. *M. smegmatis* knockouts in *mprA* (hygromycin marked) and *sigE* (zeomycin marked) were obtained from Thomas C. Zhart (Medical College of Wisconsin) and Robert Husson (Boston Children Hospital, Harvard University),

respectively, as kind gifts. The *M. tuberculosis* knockout in *mprA* was obtained from Issar Smith's laboratory (PHRI Center, Rutgers University), while a *sigE* mutant of *M. tuberculosis* were previously reported in Manganelli et al. (4).

**(ii) Construction of P*mprA*-*dnaK* fusion.** DnaK was ectopically expressed from the *mprA* promoter. For construction of the *mprA* promoter::*dnaK* fusion, DNA fragments containing sequences 470 bp upstream of *mprA* plus the first codon of the *mprA* open reading frame were PCR amplified and fused in frame with the N terminus of the *dnaK* coding region. Primers are listed in Table S1 in the supplemental material. The fusion construct was finally cloned into an integrative *E. coli*-mycobacterium shuttle vector, pMV306-kan (41, 42). Construction of the P*mprA*-*dnaK* fusion was verified by DNA sequencing. The P*mprA*-*dnaK* fusion construct was electroporated in *M. tuberculosis* and the transformants were selected on kanamycin plates. Integration at the attP locus of the genome was verified by PCR.

**SDS treatment.** For gene expression analyses, mid-log cultures of *M. tuberculosis* and *M. smegmatis* were washed prewarmed (at 37°C) 7H9 medium and treated with 0.03% and 0.02% SDS, respectively. These concentrations of detergent are bacteriostatic and had no bactericidal effect (18) (see Fig. S1 in the supplemental material and Fig. S8 in reference 18). Bacterial cultures were grown at 37°C with shaking before and after SDS treatment. Gene expression analyses were performed from two sets of assays, an SDS time course and an SDS concentration course. For time course experiments, mid-log cultures of *M. tuberculosis* and *M. smegmatis* were treated with specific bacteriostatic SDS concentrations (mentioned above). Culture aliquots (2 ml) were harvested at various time intervals up to 2 h of SDS treatment for RNA extraction. After 2 h of SDS treatment, part of the bacterial culture was centrifuged and a pellet was resuspended in SDS-free 7H9 broth (prewarmed) and incubated at 37°C with shaking. Aliquots were collected at various time intervals up to 2 h of incubation in SDS-free medium for RNA extraction.

For SDS concentration course experiments, exponentially growing cultures were treated for 2 h with increasing doses of SDS ranging from 0% to 0.03% (*M. tuberculosis*) or 0% to 0.02% (*M. smegmatis*). Aliquots were collected after 2 h of treatment for RNA extraction. After 2 h of incubation with highest SDS concentration (0.03% for *M. tuberculosis* or 0.02% for *M. smegmatis*), bacterial cultures were equally distributed in different tubes and centrifuged, and pellets were resuspended in 7H9 medium (prewarmed) with decreasing SDS concentrations ranging from 0.03% to 0% (*M. tuberculosis*) or 0.02% to 0% (*M. smegmatis*). Bacterial cultures were further incubated at 37°C with shaking for 2 h. Aliquots were collected after 2 h for RNA extraction. As a control, samples treated with the same doses of SDS before and after centrifugation were tested to avoid any experimental artifacts.

**RNA extraction and enumeration of bacterial transcripts.** Details for RNA extraction and gene expression analysis were mentioned in Datta et al. (18). Briefly, mycobacterial cells were disrupted by bead beating (Mini-BeadBeater-16; BioSpec Products, Bartlesville, OK) in the presence of 1 ml TRI reagent (Molecular Research Center, Cincinnati, OH). The aqueous phase was separated by adding 100 $\mu$l BCP reagent (Molecular Research Center, Cincinnati, OH) and was collected after centrifugation. Total RNA was precipitated with isopropanol, washed with 75% ethanol, air dried, and resuspended in diethyl pyrocarbonate (DEPC)-treated H$_2$O for storage at $-80$°C. Reverse transcription reactions were performed with ThermoScript reverse transcriptase (Invitrogen, Carlsbad, CA). Reverse transcription reactions were performed with random hexamers. Bacterial transcripts were quantitated by real-time measurements using gene-specific primers and molecular beacons. Gene copy number was normalized to the 16s rRNA copy number (43). Nucleotide sequences of PCR primers and molecular beacons are listed in Table S1 in reference 18.

**Survival analysis of wild-type and engineered P*mprA*-*dnaK* strain.** *M. tuberculosis* cultures were grown up to the exponential phase using appropriately supplemented 7H9 liquid medium. Exponentially growing cultures were treated with bacteriostatic (0.03%) and bactericidal ($>$0.03%) doses of SDS for 4 h or 24 h. Posttreatment, bacterial cultures were incubated at 37°C with shaking. Aliquots were plated for CFU enumeration before and after 4 h or 24 h of treatment. Input recovery was determined compared to CFU before the treatment.

**Computational methods. (i) MprAB-σ$^E$ network models.** The dynamic ODE model describing MprAB, $\sigma^E$, and RseA concentrations was based on previous work from our team (15). We retained most of the model components, except for two changes. First, instead of the MprB autophosphorylation rate increasing with signal, we assume that MprB exists in two conformations, kinase (MprB*) and phosphatase (MprB). We assume that only the kinase form undergoes autophosphorylation, while only the phosphatase form can dephosphorylate MprA-P. The rate of conversion from MprB to MprB* ($k_1$) was dependent on SDS concentration. Second, we introduced degradation of RseA (free or $\sigma^E$-bound) with a rate ($k_{rd}$) dependent on SDS concentration. Degradation of $\sigma^E$-RseA results in a positive flux for $\sigma^E$. Detailed transcription, translation, and posttranslational interaction reactions are described in Text S1 in the supplemental material.

Instead of a first-order activation of MprB, the DnaK model includes the following second order reaction, where MprB (kinase) binds DnaK into a complex (phosphatase).

$$\text{MprB} + \text{DnaK} \underset{k_{bdb}}{\overset{k_{bdf}}{\rightleftharpoons}} [\text{MprB·DnaK}]$$

In the presence of SDS, unfolded protein load (UF) builds up, leading to DnaK switching away from binding MprB to binding unfolded proteins:

$$UF+DnaK \underset{k_{dsb}}{\overset{k_{dsf}}{\rightleftharpoons}} [UF \cdot DnaK]$$

The total amount of UF remains constant and is dependent on SDS concentration. At $t=0$, UF amount increases from 0 to UF(SDS). Upon deactivation of stress, we assume that all UF is washed away and that all DnaK-bound UF is freed during the washing. All reactions are summarized in Text S1, which also includes ordinary differential equations and parameter tables.

**(ii) Signal dependence.** SDS concentration was incorporated into the models with a hill function. Parameters such as RseA degradation, phosphatase-to-kinase switching (canonical TCS model), and DnaK target concentration (DnaK model) were dependent on SDS with a saturating function, as shown below:

$$
\begin{aligned}
k_1 &= k_1(0) + k_{1,max}\left(\frac{SDS^{n1}}{K^{n1} + SDS^{n1}}\right) \\
k_{rd} &= k_{rd}(0) + k_{rd,max}\left(\frac{SDS^{n2}}{K_2^{n2} + SDS^{n2}}\right) \\
UF &= UF_{max}\left(\frac{SDS^{n1}}{K^{n1} + SDS^{n1}}\right)
\end{aligned}
$$

**(iii) Time course and dose-response simulations.** Input signal enters a model at two points, MprAB activation (different depending on the model) and $\sigma^E$ activation (through RseA degradation).

**(iv) Time course simulation.** The steady state of the ODE model is initially simulated at no stress. At $t=0$, both signal parameters increase to the level corresponding to high SDS concentration (0.03%). With the prestress steady state as the initial condition, the ODEs are simulated to obtain a time course of *mprA* and *sigE* mRNA between $t=0$ and 120 min (see Text S1). The values are normalized to respective prestress levels to obtain a fold change mRNA time course for stress activation. Using the state at $t=120$ min as the initial condition and setting signal parameters corresponding to 0 SDS, the deactivation time course of mRNA is obtained.

**(v) Dose-response simulation.** The above prestress initial condition is also used to numerically compute transcript levels at $t=120$ min for parameters corresponding to each intermediate SDS concentration (Fig. 3c and d). This gives us the simulated dose-response for unstressed cells. The state at $t=120$ min at high SDS concentration is used as initial condition to compute transcript levels at the same intermediate concentrations to give the simulated dose-response for previously stressed cells. All ODE solutions were obtained with the ode15s solver in MATLAB.

**(vi) Error calculation and parameter fitting.** Experimental mRNA was measured using reverse transcription-quantitative PCR (qRT-PCR) and normalized to 16S rRNA as an internal control. Each mRNA measurement (at different time points or under different treatment conditions) was then normalized to the unstressed measure to generate a fold change value. Simulated mRNA levels were also normalized to unstressed levels. Error was then calculated as a sum of squared residual of corresponding simulated and measured fold change values at the last time point, $t_f = 120$ min.

$$
\begin{aligned}
\text{fold change mRNA}(t, SDS) &= \frac{mRNA(t, SDS)}{mRNA(0,0)} \\
\text{time course error} &= \Sigma_{i=1}^{n}\left[\left(\text{fold change mRNA}[t_i, 0.03]_{sim} - \text{fold change mRNA}[t_i, 0.03]_{expt}\right)^2\right] \\
\text{dose-response error} &= \Sigma_{i=1}^{m}\left[\left(\text{fold change mRNA}[t_f, SDS_i]_{sim} - \text{fold change mRNA}[t_f, SDS_i]_{expt}\right)^2\right]
\end{aligned}
$$

Parameter fitting was performed using particle swarm optimization with MATLAB, using the above error as objective function. Many of the kinetic rate constants have not been measured experimentally in *M. tuberculosis*. Given that the number of data points is low and the number of parameters is high, a family of parameter sets was obtained for each model to account for "sloppiness" in parameters.

**(vii) Analysis of simulations.** Response time ($\tau_{90}$) was estimated as the time point after stimulus at which the simulated mRNA level was 90% of that at steady state. For experimental data, $\tau_{90}$ was estimated as the window between the latest data point at which the mRNA level is below 90% and the first data point at which it is above 90% of the final value (at 2 h post stress). Degree of hysteresis was calculated as a mean of difference between fold change mRNA levels for previously stressed and not stressed cells (red and black markers, respectively, in Fig. 3c and d) at two intermediate SDS concentration levels (0.02% and 0.025%).

## SUPPLEMENTAL MATERIAL

Supplemental material is available online only.

**TEXT S1**, PDF file, 0.2 MB.
**FIG S1**, PDF file, 0.1 MB.
**FIG S2**, PDF file, 0.1 MB.
**FIG S3**, PDF file, 0.1 MB.
**FIG S4**, PDF file, 0.1 MB.
**FIG S5**, PDF file, 0.2 MB.
**FIG S6**, PDF file, 0.2 MB.

**FIG S7**, PDF file, 0.1 MB.
**TABLE S1**, PDF file, 0.1 MB.

## ACKNOWLEDGMENTS

The research was supported by Welch Foundation grant C-1995 to O.A.I. and by NIH grants GM 096189 to O.A.I. (co-principal investigator: M.L.G.) and AI 122309, AI 104615, and HL 149450 to M.L.G.

We thank Angela Cintolesi for her contributions to the mathematical models and parameter-fitting algorithms.

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
