## [Reviewer comments · mSystems]

Chaperone-mediated stress-sensing in *Mycobacterium tuberculosis* enables fast activation and sustained response

Satyajit Rao, Pratik Datta, Maria Laura Gennaro, and Oleg Igoshin

Corresponding Author(s): Oleg Igoshin, Rice University

Review Timeline:

Submission Date:	September 23, 2020
Editorial Decision:	December 1, 2020
Revision Received:	December 5, 2020
Editorial Decision:	December 25, 2020
Revision Received:	January 3, 2021
Accepted:	January 8, 2021

Editor: Yogendra Singh

Reviewer(s): The reviewers have opted to remain anonymous.

Transaction Report:

DOI: <https://doi.org/10.1128/mSystems.00979-20>

Dear Editor and Reviewers,

We appreciate the thoughtful feedback of the two anonymous reviewers on the manuscript including careful reading and suggestions of editorial changes. We have implemented all the suggested text changes as indicated below. We have also included some additional data and model simulations that clarify the questions raised by the Reviewer #2. However, as indicated in our responses to comments #10 and #11 we opted out from performing the additional experiments suggested there. While interesting, the experiments suggested are tangential to the main flow and conclusions of the manuscript. Moreover, given that the experimental team-members of the manuscript are currently involved in time-sensitive COVID-19 research projects, waiting for completion of these experiments will lead to significant delay in publication with little bearing of the main conclusions of the work.

Reviewer #2 (Comments for the Author):

Research article

Title- "Chaperone-mediated stress-sensing in Mycobacterium tuberculosis enables fast activation and sustained response"

Comments

Using transcriptional analysis and mathematical modelling approach, the study of Rao et al. presents the mechanistic view of stress sensing by MprAB TCS and its response dynamics. The main observation of this work agrees with previously known sensing-activation mechanism and the 'hysteresis' seen in *M. tuberculosis* but not in *M. smegmatis*. Though, the authors have presented the interesting study there are certain points which need to be addressed before considering this manuscript for the publication. The following points includes both major and minor comments:

1. Please maintain uniformity while writing *M. tuberculosis* and *M. smegmatis* throughout the paper.

-We checked and revised species names as indicated in red marked-up version of the manuscript.

2. Line 21, rephrase the sentence 'confirming our proposed stress-sensing mechanism' as it should reflect the observation of the present study.

-We changed that passage to read 'confirming the role of DnaK in fast and sustained response'

3. Line 49, correct the sentence stating 'the Asp domain of MprA', it is the phosphorylation domain or Asp residue.

-We changed to Asp residue as suggested

4. Line 104-105, please provide reference for the statement or author should present the results which is used to decide the concentration of SDS used in this study.

-We now include supporting Figure (Fig S6) showing this data for *M. smegmatis* and cite prior work from our team for *M. tuberculosis*

5. Line 106, SDS exposure for 3h', however, the method section states the effective SDS exposure time used in this study is 2h.

-We changed to 2h

6. Line 110, instead of using the word 'maximal SDS' please mention the exact concentration used.

-We changed to maximal SDS=0.02%

7. Line 117-119, can we predict the hysteresis on the basis of only one factor, what if in *M. smegmatis* the transcriptional regulation of MprAB is directed by different sigma factor.

-We note that hysteresis requires a positive feedback loop. Therefore, additional sigma factor will lead to hysteresis only if that factor by itself is under MprAB control. We are not aware of such factors for *M. smegmatis* and in fact our data demonstrates the lack of hysteresis so no such factor needs to be included in the picture.

8. I would assume that the exposure of surface stressors like SDS (intermediate concentrations) will greatly affect the extra-cytoplasmic compartment (like proteins folding as well as the membrane behaviour) and not the cytoplasmic compartment. Here, I have few comments related with this description-

a. Instead of using general sentence 'the unfolding or misfolding of proteins in cells' which seems very confusing, authors should specifically write the compartment where usually the SDS exposure has maximal effect and causing the protein unfolding. Also please verify the information and correct as at few places mentioned unfolded protein accumulation while at some places it is unfolded peptide accumulation.

-We now use "unfolded/misfolded extra-cytoplasmic proteins".

b. While describing the MprB binding with DnaK, it would be clearer if authors can locate the domain in MprB where DnaK binds.

- Report by Bretl et al (2014) identifies MprB extracytoplasmic domain (ECD) as critical to suppression of signaling in unstressed condition. Additionally, they show that MprB (full length as well as ECD) immunoprecipitates DnaK. Given these findings together with their following two findings: (a) Absence of MprB (ECD) leaves the MprAB TCS constitutively active in unstressed conditions and (b) overexpression of DnaK suppresses MprAB activation even under high SDS concentrations (0.05%), we conclude that DnaK binds MprB at the extracytoplasmic domain. We change our language to clarify accordingly.

c. In the model it is not clear whether DnaK binds to MprB in the phosphatase active form or in the kinase form in unstressed condition. Here, I assume that instead of suppressing kinase activity of MprB (which should not be present in the unstressed condition) the chaperone DnaK is actually stabilizing the phosphatase conformation and not letting the SK to switch to its kinase form and give unwanted response.

-In the current implementation of the model (Text S1), we assume that binding of DnaK to MprB suppresses kinase activity and enables phosphatase activity. In other words, only DnaK-MprB catalyzes dephosphorylation reaction and only unbound form of MprB can autophosphorylate. We now explicitly mention these assumptions in the main text and depict these on the figure.

Also, a small query here, whether the membrane behaviour could also be a factor in conformational switching of MprB upon surface stress.

-While certainly plausible, such activation mechanism will be mathematically similar to the simple activation mechanism discussed in Fig. 3 and the corresponding section. Increase of MprB concentration due to the feedback unlikely to affect its interactions with the membrane and cause hysteresis.

9. Line 278, this para describes the MprA role in the unphosphorylated state. MprA is autoregulatory it can also bind to its own promoter, does the current model fit if authors include this as well?

-We thank the reviewer for pointing out the lack of this important control. We now include a supporting figure showing that inclusion of unphosphorylated MprA activation of its own promoter showing qualitatively similar results. While some residual increase in the level of *mprA* transcription is seen upon removal of SDS, the effect is much smaller than that of *sigE* and can not be ruled out from the experimental data.

10. Line 302-306, As the fig 3a and b shows fast accumulation of the transcript. Here, I would assume that, in the initial timepoint unstressed cells should show stress response, where stress will lead to activation of MprB and hence upregulation of both MprA and *sigE* along with DnaK. As overexpression of DnaK exceeds the MprB level it will cause attenuation of stress response with time. Here, authors should perform time response analysis to check this.

-We agree that performing time-response analysis is an interesting experiment but do not see how results of it would drastically affect the conclusions of the paper. Our model simulations showed that the exact nature of this

time response is sensitive to multiple unknown parameters including the level of basal DnaK expression from the activated promoter, the degree to which unphosphorylated MprA affects its target promoters etc. We therefore opted to leave this experiment for future studies.

11. Line 306-308, Here, authors should confirm this by overexpressing DnaK in the *sigE* deleted strain of *M. tuberculosis*.

-We agree that performing the experiments with the strain in which DnaK is induced in *sigE* deletion strain will allow to experimentally confirm this hypothesis. However, since this strain is not available and needs to be constructed and that this point is not significantly affects our conclusions, we opted to rephrase the text to explicitly present this hypothesis as a prediction of this model. Furthermore, we now add supporting figure that demonstrates that “induction of *mprA* is eliminated in the model in which no stress-dependent RseA degradation is present or if *sigE* is knocked out.”

12. Most of the SKs and RRs are known to functions as dimer which is not accounted in the model. How does this aspect affect the final output in the model?

-We implicitly consider both in our model. We assume that SK dimer is stable and that this is the default configuration for this species. So each MprB in our model refers to one of the model is a monomer of such dimer. Dimerization of MprA is accounted by using Hill equation with Hill coefficient of 2 to describe transcriptional activation of MprA. This is common approximation employed by us and others previously.

13. Fig S2, please label for red and blue lines

- Added legend

14. Authors should pay more attention on all the schematics used in this work. Here, I have few comments and suggestions for the same-

a. Fig 3e, 4a, the arrow direction at many points in the schematics is not proper (for

example, the arrow direction shows the MprB translated directly as active species). The translational product of the operon should be as inactive species.

- We thank the reviewer for pointing out this mistake. We have corrected it and made significant modifications to schematics of figures. We also note that SI Text contains all the chemical reactions contained in the model and their kinetics in case ambiguities arise.

b. If author has any specific reason to indicate some events with dotted line while other with solid lines, please specify.

- In our schematic, dashed lines represent transcriptional regulation. We have now indicated that in the figure caption.

c. In fig 3e and 4a, the binding and release events of sigE with RseA is not clear.

-We updated the schematic to accurately reflect SigE and RseA binding and release

d. In fig 3e and 4a, no description is given for symbol (ϕ) and red zig zag mark.

-In the schematic ϕ represents degradation products of the protein RseA and zig-zag mark represents SDS stress. We include this description in the caption.

e. Fig 4a, the schematic will be more explainable if authors can include stressors at the required points as well as to specify the route of DnaK action, as the present schematic is not satisfying the description given in the result section or in the figure legend.

We have revised the schematics to explicitly include alternative partner model with DnaK

f. Fig 4g, for better discrimination in the binding affinity one state of MprA can be presented with solid line while other with dotted line.

- We thank the reviewer for this suggestion and but, in light of adopted notation with transcriptional regulation represented by dashed line, we opted to use a lighter color and thinner arrow instead.

g. Fig 5a, the schematic is not satisfying the description given in the figure legend.

- We updated schematic to explicitly show *dnaK* synthesis from a native copy and from the copy engineered to express from the *mprA* promoter.

15. Fig 4f, the description given in the figure legend is not matching with the labelling of the graph.

- We thank the reviewer for pointing out this discrepancy and address it in the figure caption

Reviewer #3 (Comments for the Author):

Review of Rao et al for mSystems

In this paper, the authors used a combination of experimental work and modeling to better understand the regulation implicated during surface-stress response in *Mycobacterium smegmatis* and *M. tuberculosis*. They responded quite extensively to the previous reviewers' comments and modified their manuscript accordingly. Modeling being far from my specialty, I could not properly assess the validity of this parts of the paper, but I trust/hope that the previous reviewers might have been able to do so and did not find any errors.

I have only a few comments. At least the major ones should be addressed by the authors before publication. (I used the line numbers of the marked up manuscript)

Major points:

This paper seems to be a continuation of previous studies from the same lab where the authors implement new discovery from other labs to ameliorate their model and its predictive capability and confirm experimentally the obtained predictions of cellular responses. It is some time not clear what comes from their previous work or the work of others. Implementation of the previous reviewers' comments related to the DnaK hypothesis

is good but sometimes incomplete (for example: l21: "our" should be replaced by "the"). This should be checked thoroughly.

-We changed the language as indicated in #2 above and carefully checked the language throughout.

I also think that adding some possessive when the authors indeed speak about works they themselves published a few years back would make the overall manuscript clearer. For example: l131-133 = "This contrasts the predictions of the previous model of (15). To illustrate this we have simulated dose-response using the model and parameters of Tiwari et al. (Figure 2a)." This could be rephrase as: "This contrasts the predictions of our previous model in Tiwari et al. 2010 (15). To illustrate this we have simulated dose-response using this model and parameters (Figure 2a)." This kind of changes should be made throughout the manuscript (l143: "To further compare the predictions of (15) model to experimental observations," -> "To further compare the predictions of our previous model (15) to experimental observations," ; l150-151: "Notably, time course simulation of (15) model shows very slow response" -> "Notably, time course simulation of our previous model (15) shows very slow response" ; same in l164, l168, l197). The figure 2 legend should also include that the presented simulations where computed using the previous model from Tiwari et al 2010.

-We have revised all the mentions of Tiwari et al as requested.

Minor points:

- l27: "- MprAB two-component system and alternative sigma factor σ^E ": a dash should be present after σ^E , or put this between parenthesis instead

- l323: add ref

- l368: add ref

-We implemented all the changes suggested.

October 21, 2020

Prof. Oleg A Igoshin
Rice University
Bioengineering
MS142
P.O. Box 1892
Houston, Texas 77005

Re: mSystems00979-20 (Chaperone-mediated stress-sensing in *Mycobacterium tuberculosis* enables fast activation and sustained response)

Dear Prof. Oleg A Igoshin:

Dear Dr. Igoshin,
I have received two reviews for the revised version of your manuscript. One of the reviewer still has suggestions for improvement and corrections. I request you to incorporate each comment carefully in the manuscript.

Below you will find the comments of the reviewers.

To submit your modified manuscript, log onto the eJP submission site at <https://msystems.msubmit.net/cgi-bin/main.plex>. If you cannot remember your password, click the "Can't remember your password?" link and follow the instructions on the screen. Go to Author Tasks and click the appropriate manuscript title to begin the resubmission process. The information that you entered when you first submitted the paper will be displayed. Please update the information as necessary. Provide (1) point-by-point responses to the issues raised by the reviewers as file type "Response to Reviewers," not in your cover letter, and (2) a PDF file that indicates the changes from the original submission (by highlighting or underlining the changes) as file type "Marked Up Manuscript - For Review Only."

Due to the SARS-CoV-2 pandemic, our typical 60 day deadline for revisions will not be applied. I hope that you will be able to submit a revised manuscript soon, but want to reassure you that the journal will be flexible in terms of timing, particularly if experimental revisions are needed. When you are ready to resubmit, please know that our staff and Editors are working remotely and handling submissions without delay. If you do not wish to modify the manuscript and prefer to submit it to another journal, please notify me of your decision immediately so that the manuscript may be formally withdrawn from consideration by mSystems.

Sincerely,

Yogendra Singh

Editor, mSystems

Journals Department
Reviewer comments:

Reviewer #2 (Comments for the Author):

Research article

Title- "Chaperone-mediated stress-sensing in Mycobacterium tuberculosis enables fast activation and sustained response"

Comments

In Rao et al. 's revised study, some points have not been addressed correctly or, if addressed, not included in the main text. Here, I have specific issues based on the previous comments, which I feel necessary to address.

1. As suggested, the uniformity is not maintained throughout the paper while writing M. tuberculosis and M. smegmatis.
2. Lane 48, Corrected, but to be more specific, should be 'Asp residue of the receiver domain of MprA'.
3. Lane 102-104, authors response to comment 4- 'We now include supporting Figure (Fig S6) showing this data for M. smegmatis and cite prior work from our team for M. tuberculosis'. In the paper, CFU data for M. smegmatis is present in Fig S1, but no data or reference has been given for M. tuberculosis. The whole point of raising this comment is to check how these two Mycobacterium species respond (in terms of cellular growth) to different SDS concentrations, and their respective mutant strain (Δ sigE and Δ mprA) behaves similarly or not.
4. Lane 110, a new unexplained term is introduced 'reffig:exptb'.
5. line 279-288, Authors response to comment 9, Authors should mention this result in the main text.
6. Authors response from comment #10 and #11, I can understand that the current pandemic has largely affected most of the research activities. However, I am surprised by the author's response stating 'SigE deletion strain is not available and needs to be constructed' as authors have used this strain in this study. What will happen where no stress-dependent RseA degradation is present or if sigE is knocked out on DnaK overexpression and MprAB TCS activation. Fig S6 needs more

explanation in the figure legend: the triangles and what is represented by solid lines.

The description is given in the para 302-315, more especially the statement, 'MprAB TCS is not activated following SDS treatment' and ' σ E activates the mprA promoter. This does not satisfy the model given in the Fig 5a. Is it the unphosphorylated MprA or residual sigE or both, which leads to the DnaK modest upregulation? How much upregulation is sufficient to attenuate the MprAB TCS activation?

7. The authors should pay more attention while presenting the schematics. In Fig 3e, the unphosphorylated MprB indicating the phosphotransfer, while phosphorylated MprB indicates the dephosphorylation of MprA.

8. In Fig 4a, the arrow indicates the MprA-P give rise to unphosphorylated MprA and MprB.

Reviewer #3 (Comments for the Author):

The authors responded to the reviewers comments and revised the manuscript in an appropriate manner. The paper is clear and the results are interesting.

1. As suggested, the uniformity is not maintained throughout the paper while writing *M. tuberculosis* and *M. smegmatis*.

We thank the reviewer for highlighting lack of uniformity in writing species names and make changes in the Methods section.

2. Lane 48, Corrected, but to be more specific, should be 'Asp residue of the receiver domain of MprA'. (Line 48) We updated our text as suggested.

3. Lane 102-104, authors response to comment 4- 'We now include supporting Figure (Fig S6) showing this data for *M. smegmatis* and cite prior work from our team for *M. tuberculosis*'. In the paper, CFU data for *M. smegmatis* is present in Fig S1, but no data or reference has been given for *M. tuberculosis*. The whole point of raising this comment is to check how these two Mycobacterium species respond (in terms of cellular growth) to different SDS concentrations, and their respective mutant strain (Δ sigE and Δ mprA) behaves similarly or not.

In lines 102-104 we refer to Figure S1 for *M. smegmatis* growth behavior as a function of SDS, whereas for *M. tuberculosis* we cite Datta et al 2015 Mol Micro (PMID: 25899163) in lines 128-129. For clarity we now refer to the specific supplemental figure from the reference in the legend for Figure 2 and in line 129. We also note, in the Datta et al paper no differences in the survival for the first 6hrs of SDS exposure) is observed for sigE and other mutant strains.

4. Lane 110, a new unexplained term is introduced 'reffig:exptb'.

We thank the reviewer for highlighting a typographical error in our LaTeX command and we update our text to refer to the appropriate figure (Line 111).

5. line 279-288, Authors response to comment 9, Authors should mention this result in the main text.

Following the suggestion, we update our Figure S5 with simulations of the model with autoregulation mediated by the unphosphorylated MprA (dashed lines) and briefly discuss this control in the main text(Lines 284-286).

6. Authors response from comment #10 and #11, I can understand that the current pandemic has largely affected most of the research activities. However, I am surprised by the author's response stating 'SigE deletion strain is not available and needs to be constructed' as authors have used this strain in this study.

We apologize for confusion: our response to #10 and #11 indicated that we did not construct the strain engineered to overexpress DnaK with a background of sigE knockout. However, given the significantly attenuated *mprA* induction in *sigE* background (Fig. 1e), it is hard to see how further attenuating it with DnaK overexpression will be informative.

What will happen where no stress-dependent RseA degradation is present or if sigE is knocked out on DnaK overexpression and MprAB TCS activation.

As we discuss in our main text (Lines 319-321), our model predicts in case of no stress-dependent RseA degradation, the modest *mprA* upregulation seen in sigE-competent strain will be lost (Figure S6).

Fig S6 needs more explanation in the figure legend: the triangles and what is represented by solid lines.

We update the legend for Figure S6 to clarify the meanings of symbols and solid lines.

The description is given in the para 302-315, more especially the statement, 'MprAB TCS is not activated following SDS treatment' and ' σ E activates the *mprA* promoter. This does not satisfy the model given in

the Fig 5a. Is it the unphosphorylated MprA or residual sigE or both, which leads to the DnaK modest upregulation? How much upregulation is sufficient to attenuate the MprAB TCS activation?

We have now updated the graphical representation of our engineered strain (Figure 5a). The schematic now also includes the RseA-SigE pathway. We believe that this pathway is responsible for upregulation of dnaK through binding the SigE-dependent site in PmprA.

We also update the main text for clarity on the pathway through which mprA (as well as DnaK) is upregulated in this strain. We also discuss quantitative estimates on the upregulation of DnaK given our model and parameters. (Lines 316-319)

7. The authors should pay more attention while presenting the schematics. In Fig 3e, the unphosphorylated MprB indicating the phosphotransfer, while phosphorylated MprB indicates the dephosphorylation of MprA.

8. In Fig 4a, the arrow indicates the MprA-P give rise to unphosphorylated MprA and MprB.

We thank the reviewer for highlighting errors in our schematics in Figures 3 and 4. We have now updated our graphical representations of phosphotransferase and phosphatase reactions involving MprA and MprB in Figures 3,4, and 5.

December 25, 2020

Prof. Oleg A Igoshin
Rice University
Bioengineering
MS142
P.O. Box 1892
Houston, Texas 77005

Re: mSystems00979-20R1 (Chaperone-mediated stress-sensing in *Mycobacterium tuberculosis* enables fast activation and sustained response)

Dear Prof. Oleg A Igoshin:

Although, it was second revision but still there are minor mistakes. I suggest that authors should carefully go through the manuscript and remove any existing error. Once the revised version is submitted manuscript will be accepted.

Below you will find the comments of the reviewers.

To submit your modified manuscript, log onto the eJP submission site at <https://msystems.msubmit.net/cgi-bin/main.plex>. If you cannot remember your password, click the "Can't remember your password?" link and follow the instructions on the screen. Go to Author Tasks and click the appropriate manuscript title to begin the resubmission process. The information that you entered when you first submitted the paper will be displayed. Please update the information as necessary. Provide (1) point-by-point responses to the issues raised by the reviewers as file type "Response to Reviewers," not in your cover letter, and (2) a PDF file that indicates the changes from the original submission (by highlighting or underlining the changes) as file type "Marked Up Manuscript - For Review Only."

Due to the SARS-CoV-2 pandemic, our typical 60 day deadline for revisions will not be applied. I hope that you will be able to submit a revised manuscript soon, but want to reassure you that the journal will be flexible in terms of timing, particularly if experimental revisions are needed. When you are ready to resubmit, please know that our staff and Editors are working remotely and handling submissions without delay. If you do not wish to modify the manuscript and prefer to submit it to another journal, please notify me of your decision immediately so that the manuscript may be formally withdrawn from consideration by mSystems.

Sincerely,

Yogendra Singh

Editor, mSystems

Journals Department
Reviewer comments:

Reviewer #2 (Comments for the Author):

Few minor points.

1. Lane 320-322, for more clarity about the mechanism, authors should also comment on the stress-dependent perturbation in DnaK level and MprAB TCS activation when SigE network is lost.
2. The schematics in the fig 5a. wrongly representing the binding of MprB-p on MprA promoter.
3. In the method and other sections of the manuscript mention 2 hrs of SDS treatment, however, it is 3 hrs mentioned in the figure 5 legend.

We again thank the Editor and the reviewer for their attention to details and constructive suggestions to revise the manuscript. Our very minor changes and responses are described below.

1. Lane 320-322, for more clarity about the mechanism, authors should also comment on the stress-dependent perturbation in DnaK level and MprAB TCS activation when SigE network is lost.

We remain puzzled on why the reviewer suggest we address this somewhat tangential point in the text. Perhaps the confusion originates from the question of whether remaining excess of DnaK in that strain suffices to inhibit MprB. We note that in our model, there is a significant level of basal dnaK expression (in the absence of SDS or sigE) as there is significant level of basal mprAB expression. We now note that this basal level of DnaK is sufficient to inhibit activation MprB at its basal level of expression. In the absence of SigE activation, both DnaK and MprB levels remain uninduced (less than 10% increase in our simulations). We noted that in the lines 316-319 and revised Figure S6 caption to clarify what was done.

2. The schematics in the fig 5a. wrongly representing the binding of MprB-p on MprA promoter.

We thank the reviewer for highlighting this error in our schematic and we have corrected it.

3. In the method and other sections of the manuscript mention 2 hrs of SDS treatment, however, it is 3 hrs mentioned in the figure 5 legend.

We change the legend in Figure 5 to accurately reflect the protocol of 2hrs of SDS treatment.

January 8, 2021

Prof. Oleg A Igoshin
Rice University
Bioengineering
MS142
P.O. Box 1892
Houston, Texas 77005

Re: mSystems00979-20R2 (Chaperone-mediated stress-sensing in *Mycobacterium tuberculosis* enables fast activation and sustained response)

Dear Prof. Oleg A Igoshin:

Your manuscript has been accepted, and I am forwarding it to the ASM Journals Department for publication. For your reference, ASM Journals' address is given below. Before it can be scheduled for publication, your manuscript will be checked by the mSystems senior production editor, Ellie Ghatineh, to make sure that all elements meet the technical requirements for publication. She will contact you if anything needs to be revised before copyediting and production can begin. Otherwise, you will be notified when your proofs are ready to be viewed.

Sincerely,

Yogendra Singh
Editor, mSystems

Journals Department
Figure S6: Accept
Figure S5: Accept
Figure S2: Accept
Figure S7: Accept
Figure S4: Accept
Figure S3: Accept
Text S1: Accept
Figure S1: Accept